# Shedding Light on Problems with Hyperbolic Graph Learning

**Isay Katsman**  *isay.katsman@yale.edu*
*Department of Applied Mathematics*
*Yale University*

**Anna Gilbert**  *anna.gilbert@yale.edu*
*Department of Applied Mathematics*
*Department of Statistics & Data Science*
*Department of Electrical Engineering*
*Yale University*

**Reviewed on OpenReview:** *https://openreview.net/forum?id=rKAkp1f3R7*

## Abstract

Recent papers in the graph machine learning literature have introduced a number of approaches for hyperbolic representation learning. The asserted benefits are improved performance on a variety of graph tasks, node classification and link prediction included. Claims have also been made about the geometric suitability of particular hierarchical graph datasets to representation in hyperbolic space. Despite these claims, our work makes a surprising discovery: when simple Euclidean models with comparable numbers of parameters are properly trained in the same environment, in most cases, they perform as well, if not better, than all introduced hyperbolic graph representation learning models, even on graph datasets previously claimed to be the most hyperbolic (Chami et al., 2019) as measured by Gromov $\delta$-hyperbolicity (i.e., perfect trees). This observation gives rise to a simple question: how can this be? We answer this question by taking a careful look at the field of hyperbolic graph representation learning as it stands today, and find that a number of results do not diligently present baselines, make faulty modelling assumptions when constructing algorithms, and use misleading metrics to quantify geometry of graph datasets. We take a closer look at each of these three problems, elucidate the issues, perform an analysis of methods, and introduce a parametric family of benchmark datasets to ascertain the applicability of (hyperbolic) graph neural networks.

## 1 Introduction

Graph machine learning has undergone a meteoric rise in popularity over the course of the last several years. Given the ubiquity of graphs, and the demonstrated capabilities of machine learning in modern large-scale data contexts, it has been unsurprising to see widespread interest in designing machine learning models that can do predictive inference over these fundamental structures. Relevant graph tasks that researchers have focused on include link prediction (when a model must predict whether two nodes are connected or not) as well as node classification (when a model must predict one of a discrete set of classes for a given graph node).

Simultaneously, recent research from Bronstein et al. (2017) and others has brought attention to the fact that several kinds of complex data benefit from manifold considerations. In particular, Nickel & Kiela (2017) has shown that tree-like graphs (that is, graphs with a hierarchical structure) benefit from representation through hyperbolic node embeddings, in that such embeddings frequently yield lower distortion than their Euclidean analogs. This line of work is well motivated by Sarkar (2011), which first showed that trees can be embedded with arbitrarily low distortion in hyperbolic space—something that does not hold for Euclidean space.

In an attempt to capitalize on these improved non-Euclidean representations for tree-like graphs, a number of graph machine learning papers began to incorporate non-Euclidean representations into their pipelines. This

resulted in a number of highly influential papers, namely Ganea et al. (2018) and Chami et al. (2019), both of which later inspired even more related geometric graph machine learning work (Shimizu et al., 2020; Chen et al., 2021; Zhang et al., 2019; 2021). These papers had both novel and interesting aspects that attempted to maximize hyperbolicity of the model in an effort to make the best use of the data.

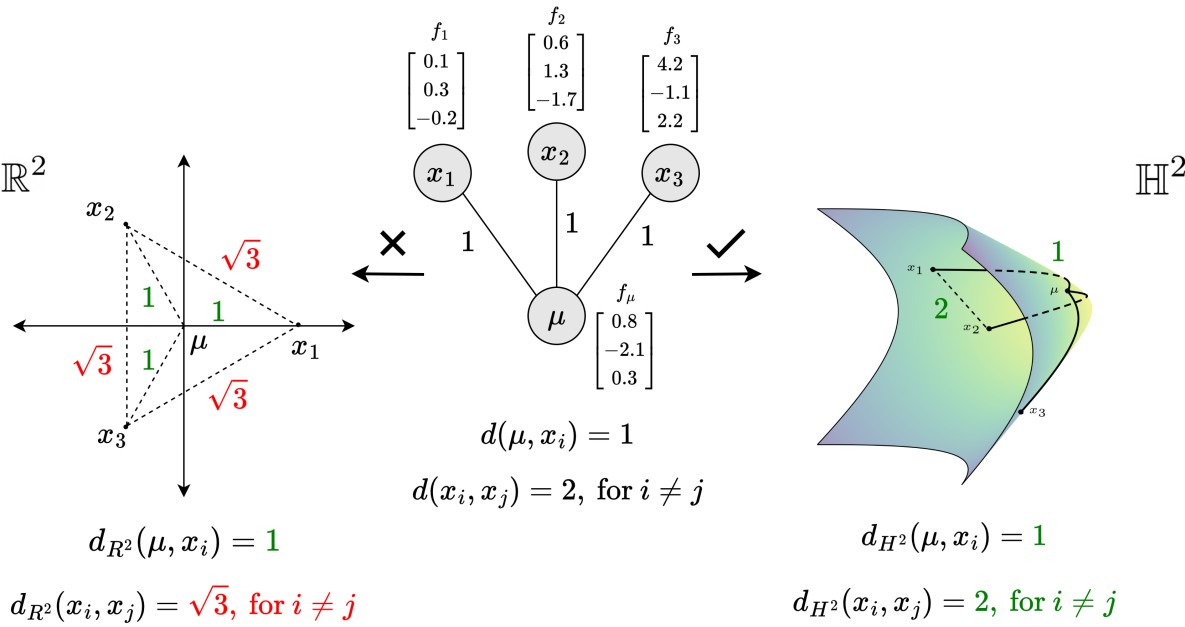

Figure 1: Above is an example where we give the best Euclidean embedding of a tripodal graph, associated with a toy graph dataset (comprised of the tripodal tree and associated node features, $f_i$), in $\mathbb{R}^2$, i.e., the embedding that has lowest average distortion. We see that the largest amount of separation we can get between the leaf nodes while maintaining the distances between the root and leaf nodes is $\sqrt{3} \approx 1.7$, falling short of 2. In contrast, we give the best hyperbolic embedding of the same tree in $\mathbb{H}^2$. Note that the embedding has high geometric fidelity, i.e., all distances can be preserved. We compute this embedding explicitly in Appendix A.

That being said, in the process of expanding and generalizing models, several papers strayed from the original theoretical and primarily geometric motivations behind non-Euclidean embeddings for trees, frequently pursuing the design of more general models without thoroughly considering the models' appropriateness for the data on which they were being used. We posit this as one of the reasons for the following main discovery of our paper: on prior benchmark tasks, when simple Euclidean models with a comparable number of parameters are trained in the same environment as these state-of-the-art hyperbolic models, the Euclidean models match or outperform the hyperbolic models on the "most hyperbolic" datasets, as measured by Gromov $\delta$-hyperbolicity. We demonstrate this specifically for the link prediction and node classification tasks presented originally in Chami et al. (2019), and used in a number of papers thereafter (Chen et al., 2021; Zhang et al., 2019; 2021). Investigating this further, we find there are three fundamental problems that led to this unfortunate state of affairs:

(i) The Euclidean baselines used in these papers were either buggy, or insufficiently trained, thereby yielding a misleading representation of inferiority.

On a more fundamental level, this led us to the observation that the original test tasks were selected incorrectly. Explicitly, we observed that the test tasks were too easy. They were easy enough to not yield statistically significant separation between the performance of Euclidean baselines and state-of-the-art hyperbolic models, they were even too weak to separate the performance of a Euclidean baseline that uses only features from the performance of a Euclidean baseline that incorporates graph

information. Most of the hyperbolic test tasks in Chami et al. (2019) are in fact trivially solvable with a simple Euclidean multi-layer perceptron that does not utilize any graph information. This suggests the need for much harder test tasks, if we are to make any conclusions about the effectiveness of incorporating graph information, let alone the effectiveness of hyperbolic methods in these contexts.

(ii) Graph machine learning models began using hyperbolic representations for Euclidean features without any justification for the hyperbolic nature of said features (in contrast to the extant hyperbolic nature of nodes in tree-like graphs). Additionally, modelling assumptions were frequently made that sacrificed geometric fidelity for the sake of convenience.

(iii) Using Gromov $\delta$-hyperbolicity (Gromov, 1987) as a proxy for graph dataset suitability to learning in hyperbolic space is flawed. Notably, $\delta$-hyperbolicity is a characteristic of the graph alone, yet a graph dataset includes node features and frequently node labels. Moreover, Gromov $\delta$-hyperbolicity is too coarse a metric for the purpose of understanding graph geometry at a finer level.

We expound on each of these three points in our paper, conducting rigorous experiments that give ample evidence to demonstrate each point, while resolving a subset of key issues and laying the grounds for a more comprehensive study. Our contributions are as follows:

1. On several of the original most hyperbolic test tasks, we demonstrate that a simple Euclidean model outperforms or matches a variety of state-of-the-art hyperbolic (and non-trivial Euclidean) models.

2. We explain the underlying causes of this phenomenon, expounding upon each of the problems (i), (ii), and (iii) above, providing relevant evidence and resolving a core subset of issues.

3. We perform an analysis of existing methods, and introduce a parametric family of benchmark datasets that help establish the applicability of (hyperbolic) graph neural networks.

## 2 Related Work

Our work addresses and is related to a specific subset of the hyperbolic machine learning (Bronstein et al., 2017) literature that deals with graphs in the context of intragraph prediction. Specifically, we deal with tasks such as link prediction, in which one must predict whether two nodes are connected (Kipf & Welling, 2017), and node classification, in which one must predict one of a set of classes for individual nodes in a graph (Kipf & Welling, 2017). That being said, we believe several problems we point out about the way hyperbolic graph learning is being done currently may hold more generally.

**Distortion** Distortion is fundamental to the discussion of geometric representation learning but is not used in a number of the aforementioned papers. One must recall the original theoretical motivation from Sarkar (2011) was to provide a natural embedding for trees (undirected, connected, acyclic graphs) in which distortion of graph distances could be made arbitrarily small. That is, we seek an embedding of the graph into a metric space $(\mathcal{M}, d_{\mathcal{M}})$ in which the distance metric $d_{\mathcal{M}}$ captures the original graph distances as well as possible. Typically, the continuous nature of $\mathcal{M}$ yields considerable benefits over the original discrete graph structure.

For a given dimension, it may be impossible to obtain a Euclidean embedding with high geometric fidelity but this may be possible in a hyperbolic space. Take for example the task of a two-dimensional embedding of the tripodal tree, with one root and three children, shown in Figure 1. The optimal embedding in $\mathbb{R}^2$ does not have high geometric fidelity as it does not preserve the graph distances faithfully. In contrast, we see in the same figure that when the same graph is embedded in $\mathbb{H}^2$, we can recover distances of 2 between the children (see Appendix A for full details), staying faithful to the original graph. This example provides some intuition for the suitability of hyperbolic space for tree embeddings.

**Graph neural networks** Graph neural networks, and perhaps more broadly, the application of modern machine learning to graph tasks, begins with the paper Kipf & Welling (2017), which introduced Graph Convolutional Networks (GCNs). The ubiquity of graphs and the power of large scale modern machine learning quickly transformed graph machine learning into an incredibly large and diverse field

(Velickovic et al., 2018). That being said, these methods were not without their problems. Wu et al. (2019) pointed out that many of the modifications being made to graph neural networks (GNNs) were at their core, superficial, and presented a way to considerably simplify GNNs. Additionally, Huang et al. (2020) showed that a very simple graph neural network construction with few parameters could outperform most state-of-the-art models with an order of magnitude more parameters. Such papers proved very useful at forcing the field to step back and think about what methodological changes were truly useful. In a similar sense, our paper aims to push current hyperbolic machine learning papers, particularly in graph contexts, for more rigorous research and careful methodological proposals.

**Hyperbolic graph machine learning** Hyperbolic graph machine learning can be viewed either as a subfield of geometric machine learning (Bronstein et al., 2017) or as a subfield of graph machine learning. Either way, it has enjoyed increasing popularity over the course of the last several years. The original line of work from Nickel & Kiela (2017) was principled in that it wanted to generate better embeddings for trees (WordNet (Fellbaum, 2000) hierarchies, to be precise) via hyperbolic embedding. Concurrently, Ganea et al. (2018) began to extend classical Euclidean neural networks to hyperbolic space. Issues in the principled nature of the work began to arise when researchers began to push hyperbolic neural networks to do intragraph prediction. A prominent early paper exemplifying this is Chami et al. (2019). While mentioning distortion several times for motivation, it loses sight of what it actually means to minimize distortion (i.e., distortion is with respect to a graph embedding). In this work, features are mapped into hyperbolic space despite the fact that there is no direct evidence for their hyperbolicity, in contrast to the nodes themselves that are best suited to hyperbolic space assuming the graph is a tree. Further papers that build upon Chami et al. (2019), such as Zhang et al. (2019), Zhang et al. (2021), and Chen et al. (2021) follow the framework of initially mapping the features into hyperbolic space. Their innovations are primarily architectural: Zhang et al. (2019) introduces Hyperbolic Graph Attention Networks (HATs), Zhang et al. (2021) introduces Lorentzian Graph Convolutional Networks (LGCNs), and Chen et al. (2021) introduces a hyperbolic neural network capable of expressively capturing Lorentz boosts, in contrast to Ganea et al. (2018). Katsman et al. (2023) introduces a general manifold version of a residual neural network and uses hyperbolic space as a use case, comparing against the above methods.

**Graph curvature** One of the claims we put forth is that geometric graph measures previously used are insufficient to capture the geometry of the graph, let alone the graph dataset (comprised of the graph, features, and frequently node labels). Notably, Gromov $\delta$-hyperbolicity has been previously used in Chami et al. (2019); Chen et al. (2021); in particular, the same number, 0, is assigned to all trees despite considerable potential differences in geometry (e.g. branch factor). Graph curvature measures, such as Ollivier-Ricci curvature (Lin et al., 2011), have been previously studied as a way to more granularly characterize graphs, but are seldom used in this context. Please see Appendix C for further details.

## 3  Background

We give the primary background necessary to understand the key results of this paper. In particular, we introduce some essential definitions from Riemannian geometry, give necessary background on hyperbolic space, and demonstrate explicitly how Chami et al. (2019); Liu et al. (2019); Zhang et al. (2019; 2021) map features from Euclidean to hyperbolic space; we also define Gromov $\delta$-hyperbolicity.

### 3.1  Riemannian Geometry

We establish relevant definitions from Riemannian geometry (Lee, 1997).

**Manifold** An $n$-dimensional manifold $\mathcal{M}$ is a topological space that is locally homeomorphic[1] to $\mathbb{R}^n$.

**Tangent space** The tangent space $T_x\mathcal{M}$ at $x$ of a manifold $\mathcal{M}$ is defined as the vector space of all tangent vectors at $x$ and is isomorphic to $\mathbb{R}^n$.

**Riemannian manifold** A Riemannian manifold is a manifold $\mathcal{M}$ equipped with a Riemannian metric, $\rho = (\rho_x)_{x \in \mathcal{M}}$, a smooth collection of inner products $\rho_x : T_x\mathcal{M} \times T_x\mathcal{M} \to \mathbb{R}$ for every $x \in M$.

---

[1]A homeomorphism is a continuous bijection with continuous inverse.

**Geodesics and induced distance function** Given a Riemannian manifold $(\mathcal{M}, \rho)$ and a curve $\gamma : [a, b] \to \mathcal{M}$, we define the length of $\gamma$ to be $L(\gamma) = \int_a^b ||\gamma'(y)||_\rho \, dt$. For $x, y \in \mathcal{M}$, the distance $d(x, y) = \inf L(\gamma)$, where $\gamma$ is any curve such that $\gamma(a) = x, \gamma(b) = y$. A geodesic $\gamma_{xy}$ from $x$ to $y$ should be thought of as a curve that minimizes this length.

**Riemannian exponential map** For each point $x \in \mathcal{M}$ and vector $v \in T_x\mathcal{M}$, there exists a unique geodesic $\gamma : [0, 1] \to \mathcal{M}$, where $\gamma(0) = x, \gamma'(0) = v$. The exponential map $\exp_x : T_x\mathcal{M} \to \mathcal{M}$ is defined by $\exp_x(v) = \gamma(1)$.

### 3.2 Hyperbolic Space

We give some necessary definitions regarding hyperbolic space. A more detailed treatment can be found in Lee (1997).

**Hyperbolic space** Hyperbolic space is the Riemannian manifold of constant negative sectional curvature $K < 0$. Here we deal with the hyperboloid model of hyperbolic space: $\mathbb{H}_K^n = \{x \in \mathbb{R}^{n+1} : \langle x, x \rangle_{\mathcal{L}} = \frac{1}{K}\}$, where $\langle x, x \rangle_{\mathcal{L}}$ is the hyperbolic inner product:

$$\langle x, y \rangle_{\mathcal{L}} = -x_0 y_0 + x_1 y_1 + x_2 y_2 + \cdots + x_n y_n$$

for $x, y \in \mathbb{H}_K^n$. It is equipped with the pullback metric from $\mathbb{R}^{n+1}$ (Lee, 1997).

**Hyperbolic exponential map** The Riemannian exponential map for the hyperboloid model has a closed form given by:

$$\exp_x^K(v) = \cosh(\sqrt{|K|}||v||_{\mathcal{L}})x + v\frac{\sinh(\sqrt{|K|}||v||_{\mathcal{L}})}{\sqrt{|K|}||v||_{\mathcal{L}}}$$

where $x \in \mathbb{H}_K^n$ and $v \in T_x\mathbb{H}_K^n$.

### 3.3 Mapping Features from Euclidean to Hyperbolic Space

Hyperbolic graph machine learning models are usually trained on graph datasets that have Euclidean features. In order for the models to process these features, they map them to the hyperboloid via the exponential map. This is the procedure followed by Chami et al. (2019); Liu et al. (2019); Zhang et al. (2019; 2021); Chen et al. (2021). Explicitly, the procedure is as follows. Let $x^E \in \mathbb{R}^n$ denote the input Euclidean features. Let $o = \{\sqrt{K}, 0, \ldots, 0\} \in \mathbb{H}_K^n$ denote the north pole (origin) in $\mathbb{H}_K^n$, which is the reference point used to perform tangent space operations. They interpret $(0, x^E)$ as a point in $T_o\mathbb{H}_K^n$ and use the exponential map to map it to $\mathbb{H}_K^n$:

$$x^H = \exp_o^K((0, x^E))$$

$$= \left( \sqrt{K}\cosh\left(\frac{||x^E||_2}{\sqrt{K}}\right), \sqrt{K}\sinh\left(\frac{||x^E||_2}{\sqrt{K}}\right)\frac{x^E}{||x^E||_2} \right)$$

Notice how the origin for exponentiation is chosen arbitrarily. Moreover, observe that this use of hyperbolic space to represent the features is separate from any notion of improved distortion of graph embedding, discussed at length in Section 2. To give a concrete example, this would map the features $f_i$ from Figure 1 into hyperbolic space, instead of producing embeddings of the graph nodes. One can perhaps posit that the features are in some way representative of the original hierarchical relationship between the nodes, but this need not be the case. In other words, this ad hoc exponentiation is not well-motivated.

### 3.4 Gromov $\delta$-Hyperbolicity

Let $G = (V, E)$ be a connected graph with distance function $d$ defined as the number of edges on a shortest path between a pair of vertices. Formally, the Gromov $\delta$-hyperbolicity can be defined using the following four-point condition (Gromov, 1987).

Given a graph $G = (V, E)$ and four vertices $x, y, u, v \in V$ with:

$$d(x, y) + d(u, v) \geq d(x, u) + d(y, v) \geq d(x, v) + d(y, u)$$

the hyperbolicity of the quadruple $x, y, u, v$ denoted as $\delta(x, y, u, v)$ is defined as:

$$\delta(x, y, u, v) = (d(x, y) + d(u, v) - (d(x, u) + d(y, v)))/2$$

and the $\delta$-hyperbolicity of the graph is $\delta(G) = \sup_{x,y,u,v \in V} \delta(x, y, u, v)$. Note that Gromov $\delta$-hyperbolicity is nonnegative and can be thought of as measuring how "tree-like" a graph is. The closer it is to 0, the closer a graph is to being a perfect tree. We reference the interested reader to Appendix B for additional intuition.

## 4    Elucidating Problems

In this section, we revisit the problems mentioned in Section 1 and provide relevant evidence for our claims.

### 4.1    Misleading Presentation of Strength of Hyperbolic Graph Models

**Properly-tuned Euclidean models outperform hyperbolic models** A somewhat persistent issue across a number of prominent geometric graph machine learning papers is either buggy or poor implementation of Euclidean baselines. Such an issue does indeed arise in the publicly available code for Chami et al. (2019). We discuss the details of the bug and the fix in Appendix F. In short, the central issue is that two lines in the implementation of the MLP normalize all Euclidean features to lie within the unit ball; commenting these two lines alleviates this issue. Although this introduces some numerical instability to the Fermi-Dirac decoder, there is also a simple one line fix for this that we also give in Appendix F. The removal of this bug allows for the Euclidean representation space to be fully utilized, i.e., the features are no longer constrained to a unit ball.

If we fix bugs in a Euclidean baseline of Chami et al. (2019), said model outperforms or matches all hyperbolic models on nearly all of the most hyperbolic datasets of that paper. The results for the non-buggy version of the multi-layer perceptron Euclidean model in Chami et al. (2019) are given in the "MLP (debugged)" row of Table 1. All results are given by mean and standard deviation over 5 trials. As is evident, the debugged Euclidean model trivially solves nearly all of the most hyperbolic tasks. The Euclidean MLP presented in the original GitHub repository[2] attains $98.7 \pm 0.2$ test ROC AUC for link prediction (vs. $72.6 \pm 0.6$ for the original) on Disease (Chami et al., 2019) and $80.3 \pm 0.7$ test F1 score for node classification (vs. $28.8 \pm 2.5$ for the original) on the same dataset (as shown in Table 1 above). These are increases of 43.5 and 20.6 standard deviations, by the standards of the originally presented link prediction and node classification results for this model, respectively. The link prediction result for Disease-M is $99.1 \pm 0.3$ test ROC AUC (vs. $55.3 \pm 0.5$ for the original) after the correction, up 87.6 standard deviations from the original result. The one exception is Disease NC, for which the difference between the debugged and original version is staggering ($80.3 \pm 0.7$ versus $28.8 \pm 2.5$), though the result is not state-of-the-art. It is also worth noting that the Euclidean MLP attains these results without the graph, since it uses features alone! The graph neural network baselines on the table (under subheading "GNN") are standard near state-of-the-art baselines used in the graph neural network literature. The "Hyp NN" models are all various different hyperbolic neural network models, with either HyboNet (Chen et al., 2021) or G-RResNet Horo (Katsman et al., 2023) being the current state-of-the-art on these test tasks.

It should be noted that a number of papers (e.g. HyboNet (Chen et al., 2021)) make direct use of the experimental setup provided originally in Chami et al. (2019), hence we effectively have that the original issues with baselines cascade and affect the experimental setup for many papers at once. To the best of our knowledge, none of these papers noticed the issues with the Euclidean MLP baseline. The experimental details for Table 1 are given in Appendix G.

---

[2]Please find the original GitHub repository for Chami et al. (2019) at https://github.com/HazyResearch/hgcn.
[3]The authors were unable to provide the dataset to obtain this result.

| | Dataset
Hyperbolicity | Disease
$\delta = 0$ | | Disease-M
$\delta = 0$ | | Airport
$\delta = 1$ | |
|---|---|---|---|---|---|---|---|
| | Task | LP | NC | LP | NC | LP | NC |
| Shallow | Euc (Chami et al., 2019) | $59.8_{\pm 2.0}$ | $32.5_{\pm 1.1}$ | – | – | $92.0_{\pm 0.0}$ | $60.9_{\pm 3.4}$ |
| | Hyp (Nickel & Kiela, 2017) | $63.5_{\pm 0.6}$ | $45.5_{\pm 3.3}$ | – | – | $94.5_{\pm 0.0}$ | $70.2_{\pm 0.1}$ |
| | Euc-Mixed | $49.6_{\pm 1.1}$ | $35.2_{\pm 3.4}$ | – | – | $91.5_{\pm 0.1}$ | $68.3_{\pm 2.3}$ |
| | Hyp-Mixed | $55.1_{\pm 1.3}$ | $56.9_{\pm 1.5}$ | – | – | $93.3_{\pm 0.0}$ | $69.6_{\pm 0.1}$ |
| NN | MLP (original) | $72.6_{\pm 0.6}$ | $28.8_{\pm 2.5}$ | $55.3_{\pm 0.5}$ | $55.9_{\pm 0.3}$ | $89.8_{\pm 0.5}$ | $68.6_{\pm 0.6}$ |
| | MLP (debugged) | $\mathbf{98.7}_{\pm 0.2}$ | $80.3_{\pm 0.7}$ | $\mathbf{99.1}_{\pm 0.3}$ | –[3] | $96.4_{\pm 0.1}$ | $95.9_{\pm 0.8}$ |
| GNN | GCN (Kipf & Welling, 2017) | $64.7_{\pm 0.5}$ | $69.7_{\pm 0.4}$ | $66.0_{\pm 0.8}$ | $59.4_{\pm 3.4}$ | $89.3_{\pm 0.4}$ | $81.4_{\pm 0.6}$ |
| | GAT (Velickovic et al., 2018) | $69.8_{\pm 0.3}$ | $70.4_{\pm 0.4}$ | $69.5_{\pm 0.4}$ | $62.5_{\pm 0.7}$ | $90.5_{\pm 0.3}$ | $81.5_{\pm 0.3}$ |
| | SAGE (Hamilton et al., 2017) | $65.9_{\pm 0.3}$ | $69.1_{\pm 0.6}$ | $67.4_{\pm 0.5}$ | $61.3_{\pm 0.4}$ | $90.4_{\pm 0.5}$ | $82.1_{\pm 0.5}$ |
| | SGC (Wu et al., 2019) | $65.1_{\pm 0.2}$ | $69.5_{\pm 0.2}$ | $66.2_{\pm 0.2}$ | $60.5_{\pm 0.3}$ | $89.8_{\pm 0.3}$ | $80.6_{\pm 0.1}$ |
| Hyp NN | HNN (Ganea et al., 2018) | $75.1_{\pm 0.3}$ | $41.0_{\pm 1.8}$ | $60.9_{\pm 0.4}$ | $56.2_{\pm 0.3}$ | $90.8_{\pm 0.2}$ | $80.5_{\pm 0.5}$ |
| | HGCN (Chami et al., 2019) | $90.8_{\pm 0.3}$ | $74.5_{\pm 0.9}$ | $78.1_{\pm 0.4}$ | $\mathbf{72.2}_{\pm 0.5}$ | $96.4_{\pm 0.1}$ | $90.6_{\pm 0.2}$ |
| | HAT (Zhang et al., 2019) | $91.8_{\pm 0.5}$ | $83.6_{\pm 0.9}$ | – | – | – | – |
| | LGCN (Zhang et al., 2021) | $96.6_{\pm 0.6}$ | $84.4_{\pm 0.8}$ | – | – | $96.0_{\pm 0.6}$ | $90.9_{\pm 1.7}$ |
| | HyboNet (Chen et al., 2021) | $96.8_{\pm 0.4}$ | $\mathbf{96.0}_{\pm 1.0}$ | – | – | $\mathbf{97.3}_{\pm 0.3}$ | $90.9_{\pm 1.4}$ |
| | G-RResNet Horo (Katsman et al., 2023) | $\mathbf{98.4}_{\pm 0.3}$ | $\mathbf{95.4}_{\pm 1.0}$ | – | – | $95.2_{\pm 0.1}$ | $\mathbf{97.4}_{\pm 0.1}$ |

Table 1: Above we give graph task results for a debugged version of the Euclidean model from Chami et al. (2019) compared to other models, including the current state-of-the-art models from Chen et al. (2021) and Katsman et al. (2023). The metrics reported are test ROC AUC for link prediction and test F1 score for node classification. Means and standard deviations are given over 5 trials. After the bug fix, the Euclidean "MLP (debugged)" entry above attains or nearly matches state-of-the-art results on most of the hyperbolic test tasks in the paper. We highlight the best result only if the result gives a p-value less than 0.01 after running a paired significance t-test against the next best result.

**Baselines for hyperbolic knowledge graph tasks** It should be noted that Euclidean baselines in other papers can frequently be made stronger. We show a more typical scenario via a subset of the knowledge graph tasks in Chami et al. (2020), specifically those that "exhibit hierarchical structures." In particular, we note that carefully tuned Euclidean models considerably decrease the gap between the baseline Euclidean models in that paper and the introduced hyperbolic models. Results are given in Appendix D (Table 3).

## 4.2 How Did this Happen?

**Poor dataset selection** The results in Table 1 suggest that the graph structure, and perhaps geometric fidelity more broadly, is irrelevant to solving node classification and link prediction on the "most hyperbolic" tasks presented in the cited subset of hyperbolic machine learning papers. Please note that both Disease and Disease-M are the most hyperbolic datasets presented, as measured by Gromov $\delta$-hyperbolicity (the $\delta$-hyperbolicity is 0, meaning the graphs from both datasets are perfect trees). The fact that a Euclidean model with no graph information (i.e. a model that only uses features) effectively solves the most hyperbolic tasks led us to recognize what is perhaps an even more fundamental problem of the original work: the graph tasks selected were quite poor, in that they could not serve their purpose of distinguishing the hyperbolic models from baseline Euclidean models, simply because the multi-layer perceptron Euclidean model already solved the tasks with no graph information. To avoid this, we claim one must first ensure that the graph tasks benefit from geometry, i.e., that usage of the graph markedly improves performance. We elaborate on this further in Section 5.

**Poorly motivated model design** A model design feature that has persisted in hyperbolic graph neural networks due to very early adoption from Kipf & Welling (2017) is the failure to distinguish between nodes and node features. Graph convolution convolves the node features as de facto substitutes for the nodes themselves. However, when we deal with these graphs in a geometric context, it is important to recall that

the benefit we obtain from Sarkar (2011) in terms of embedding trees in hyperbolic space only holds if we explicitly embed the nodes in hyperbolic space with respect to the original graph distances. Simply mapping the node features to hyperbolic space, as discussed in Section 3.3, does not necessarily capture the original graph geometry. One may perhaps make the argument that the features should by proxy capture some of the original hyperbolicity of the graph, but there are myriad counterexamples[4] for this and hyperbolicity of features has never been defined, let alone demonstrated. The failure to recognize and/or to explicitly address this is what makes the decision to exponentiate features in Chami et al. (2019); Zhang et al. (2019; 2021); Liu et al. (2019) and derivative work not methodologically principled. The construction and the theoretical motivation presented originally by Sarkar (2011) diverge.

**Poor geometric characterization of graph datasets** Another problem is that the metric used to quantify the "hyperbolicity" of the so-called "most hyperbolic" graph datasets characterizes only the graph, and moreover, does so in rather a coarse way. As we saw above, characterizing graph datasets via the graph alone can be extremely misleading, simply because the features alone are often enough to solve the relevant tasks. A proper characterization should take into account both the graph and associated node features. Additionally, using Gromov $\delta$-hyperbolicty is quite a coarse characterization of the graph, since $\delta = 0$ for all trees, regardless of considerable differences in geometry (e.g. differing branch factor). One must use more granular graph curvature metrics, for example Ollivier-Ricci curvature, to more precisely characterize graph geometry. We refer the interested reader to Appendix B for a definition of Ollivier-Ricci curvature together with intuition. That being said, Ollivier-Ricci is still a graph-level characterization and does not incorporate analysis of node features, which in some sense is a more central and pressing issue. One would need to develop a more fundamental set of tools to properly characterize node feature geometry together with the geometry of the graph; we believe this is a very fruitful and important avenue for future work.

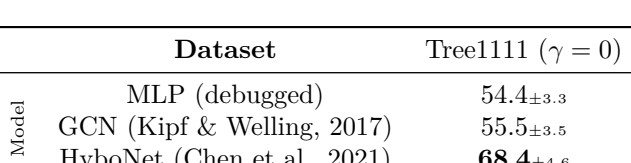

|  | **Dataset** | Tree1111 ($\gamma = 0$) |
|---|---|---|
| Model | MLP (debugged) | $54.4_{\pm 3.3}$ |
|  | GCN (Kipf & Welling, 2017) | $55.5_{\pm 3.5}$ |
|  | HyboNet (Chen et al., 2021) | $\mathbf{68.4}_{\pm 4.6}$ |

Table 2: Link prediction results (test ROC AUC) for our synthesized Tree1111 dataset, meant to illustrate a context in which Euclidean models fail.

Figure 2: Introducing parental dependence in node features very quickly leads the MLP to solve link prediction. Each data point is obtained by tuning the MLP on the relevant synthetic dataset. The average of 5 trials is reported and the error region specifies one standard deviation.

## 5 Initial Solutions

In response to the above presented problems, our solutions are to test a well-tuned graph-less Euclidean MLP baseline on a given graph dataset before making use of a graph machine learning method, using synthetic datasets to benchmark graph methods (or selecting tasks where graph structure is actually important for the task at hand, i.e., where using the features alone is not sufficient to solve the task), designing models well, based on theory (Sarkar, 2011) (which implies usefulness of graph positional embeddings), and lastly, using

---

[4]Take for example a dataset of airports for which separate nodes, say major airports in New York City and Shanghai, may be very different and far apart in the graph while being very close in feature space (e.g. similar land area, occupancy, etc.). The point more generally is that there is no guarantee that the feature space structure will mimic the graph structure.

quantitative graph dataset characterizations that capture both the graph and associated graph features to determine suitability of a graph dataset to a geometric machine learning method.

**Euclidean MLP baseline** Crucially, we recommend a well-tuned Euclidean MLP be applied first to a graph dataset (i.e. the features alone); if this method is insufficient, one should seek to use a graph neural network that makes use of the graph information together with the features. Even in this case, one should still be careful, since more recently Bechler-Speicher et al. (2024) points out different graph neural networks can perform quite differently on various benchmarks and it may be the case that the graph is not necessary (perhaps even detrimental) for a given task using a certain graph neural network.

**Synthetic graph benchmark datasets** The results in Table 1 show that these tasks make for quite a poor proxy to measure the benefit of using the graph, let alone the geometric benefit of the proposed hyperbolic approach. For proper benchmarking, one must attempt to select tasks that necessitate graph information.

**Defining Tree**$(b, \ell, \gamma, \delta, \mathcal{D})$ For the purpose of illustration, and to introduce an important set of synthetic graph machine learning benchmark tasks, we describe designing better alternative datasets to Disease and Disease-M (Chami et al., 2019). We do this by introducing a parametric family of dataset distributions that we denote by $\text{Tree}(b, \ell, \gamma, \delta, \mathcal{D})$; drawing a sample from a given dataset distribution implies generating a graph dataset whose graph is an $\ell$-level tree with branch factor $b$ and whose features are generated procedurally, level-by-level, in the following way:

$$x^{(0)} \sim \mathcal{D}$$
$$x^{(n)} \leftarrow \gamma \cdot p(x^{(n)}) + \delta \cdot v, \text{ where } v \sim \mathcal{D}$$

where $x^{(0)}$ is the root node's feature vector and $p(x^{(n)})$ is the feature vector of the parent node of $x^{(n)}$. For the sake of concreteness, we will fix $\mathcal{D} = \mathcal{N}(\mathbf{0}, I_{1000})$ (i.e. the multivariate normal distribution over $\mathbb{R}^{1000}$ with mean $\mathbf{0}$ and identity covariance matrix) and $\delta = 1$. Now note that $\gamma$ controls the level of parental dependence, which in effect controls the extent to which the features carry the graph structure (i.e. the extent to which the relationships of the Euclidean distances between features mimic the relationships of the graph distances between the corresponding nodes). If $\gamma = 0$, the features are entirely i.i.d. and carry no graph structure. To perform link prediction well on such a dataset, a graph machine learning method must make explicit use of the graph.

**Tree1111 dataset** For illustration (and as an initial starting point), we sample from $\text{Tree}(10, 3, 0, 1, \mathcal{N}(\mathbf{0}, I_{1000}))$ and call this sampled dataset Tree1111. That is, we take a graph with high branch factor and synthesize features simply by drawing them i.i.d. from the specified Gaussian; the graph and these features together comprise the dataset, which is suitable for link prediction. Note that the features contain no graph information, thereby forcing use of the graph for nontrivial performance. Moreover, the larger branch factor (10) relative to Disease (Chami et al., 2019) makes this dataset a far worse fit for Euclidean graph machine learning models. We summarize the performance for three key models on this dataset in Table 2. In particular, the Euclidean multi-layer perceptron that trivially solves Disease in Table 1 only obtains $54.4 \pm 3.3$ test ROC AUC for Tree1111 link prediction after thorough tuning. Moreover, adding graph information via the traditional Euclidean GCN also does not help, yielding a result of $55.5 \pm 3.5$ test ROC AUC. Finally, we note that the state-of-the-art hyperbolic model HyboNet (Chen et al., 2021) obtains $68.4 \pm 4.6$ test ROC AUC, a considerable improvement over both Euclidean models. This Tree1111 result is significant, in that it demonstrates a situation where graph information is not only relevant, but the straightforward Euclidean GCN treatment fails, and a hyperbolic treatment (of the graph) truly helps yield better results.

**Analysis of existing methods** Moreover, we note that when the features are completely uninformative, a number of standard Euclidean graph models do not use graph information in a more complex way than a simple GCN (when performing a per-layer analysis at initialization); the full details of this analysis are given in Appendix E. This highlights the overreliance of existing graph machine learning methods on the adjacency matrix alone, and indicates that more explicit usage of the graph information via embeddings can yield considerable benefit in situations where the graph is important.

**Tree1111$_\gamma$ datasets** To further illustrate this, as well as how when graph information is leaked into the features, graph machine learning can quickly becomes entirely unnecessary, we vary $\gamma$ from 0 to 1 in increments of 0.2, sampling a graph dataset from each parametric distribution. This produces six datasets, that we call

$\{\text{Tree1111}_\gamma | \gamma \in \{0.2k | k \in 0 \cup [5]\}\}$. We tune a Euclidean MLP on these datasets and give the results in Figure 2. As is clearly visible, as soon as $\gamma \geq 0.4$, the MLP trivially solves the link prediction task with features alone. To produce more challenging benchmark datasets, we also generate $\text{Tree1111}_{0.05}, \text{Tree1111}_{0.1}, \text{Tree1111}_{0.15}$; as can be seen from Figure 2, these datasets are not as trivially solved by the MLP and thus are reasonable additional benchmarks for hyperbolic graph machine learning models.

Note that the feature synthesis process for $\gamma > 0$ can be viewed as sampling a positional embedding for the nodes of the graph, which we see here, produces a considerable benefit and is a potential solution (yielding complementary benefit) to the fact that many graph methods do not make use of graph information in a more nuanced manner.

## 6 Conclusion

In this work, we have demonstrated that state-of-the-art geometric graph learning results are misleading, in that, for most of the hyperbolic tasks in Chami et al. (2019), a simple Euclidean model can outperform or match state-of-the-art models while using the features alone. We explain the problems that led to this state of affairs. Namely, that hyperbolic graph machine learning papers (i) suffer from buggy or weak Euclidean baselines, (ii) select test tasks that can be solved with the features alone, (iii) stray from the theoretical motivation behind hyperbolic embedding for trees, and (iv) use a graph measure ($\delta$-hyperbolicity) to characterize an entire graph dataset, as opposed to characterizing the features jointly with the graph (moreover, the graph measure used is coarse).

**Limitations** We present initial solutions in Section 5 as a substantial first step towards resolving the aforementioned problems, but recognize that we fall short of a complete understanding of all aspects.

**Future work** We resolve the first three of these four problems above. Additional work on the third problem would consist of finding real world benchmarks to match the synthetic benchmarks we introduced in this paper. The largest remaining amount of work is to be done on the fourth problem, that of introducing a metric to characterize graph datasets (that is, the graph and the associated node features). For this, one will have to characterize the interaction of the node features and the graph jointly, and may have to use more granular metrics than Gromov $\delta$-hyperbolicity (e.g. Ollivier-Ricci curvature) to gain a better geometric understanding of the underlying structure.

**Impact statement** This paper, among other things, demonstrates the need for greater care when dealing with baseline evaluation in the context of hyperbolic machine learning. We do not have many concerns about negative societal impact, and hope our results will improve the quality of evaluation in the geometric graph machine learning literature, thereby improving the overall quality of work in the field. It should be noted that we do make use of a synthetic dataset to demonstrate some of our core claims and note that we do not want to push the field entirely to the extreme of only synthetic benchmarks; ideally, we would use relevant practical real-world datasets for evaluation and finding such test tasks should be a priority for the field.

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

# Appendix

## A Hyperbolic Embedding Example Computation

In this section we specify some fundamental Riemannian constructs in the context of the hyperboloid model of hyperbolic space (Lee, 1997), use them to compute an $\mathbb{H}^2$ embedding of the tripodal graph in Figure 1, and show that the distances can be preserved with arbitrarily low distortion in hyperbolic space. We will deal with the hyperboloid model of hyperbolic space with curvature $K < 0$; the relevant point set is $\mathbb{H}^n_K = \{x \in \mathbb{R}^{n+1} : \langle x, x \rangle_{\mathcal{L}} = \frac{1}{K}\}$, where $\langle x, x \rangle_{\mathcal{L}}$ is the hyperbolic inner product:

$$\langle x, y \rangle_{\mathcal{L}} = -x_0 y_0 + x_1 y_1 + x_2 y_2 + \cdots + x_n y_n$$

for $x, y \in \mathbb{H}^n_K$. Referencing Lou et al. (2020), we note that the hyperbolic distance in this model is given by:

$$d^K_{\mathbb{H}}(x, y) = \frac{1}{\sqrt{|K|}} \cosh^{-1}(K \langle x, y \rangle_{\mathcal{L}})$$

and the Riemannian exponential map has a closed form given by:

$$\exp^K_x(v) = \cosh(\sqrt{|K|} \|v\|_{\mathcal{L}}) x + v \frac{\sinh(\sqrt{|K|} \|v\|_{\mathcal{L}})}{\sqrt{|K|} \|v\|_{\mathcal{L}}}$$

where $x \in \mathbb{H}^n_K$ and $v \in T_x \mathbb{H}^n_K$. Recall we seek a low distortion embedding for the tripodal graph shown in Figure 1 with distances $d(\mu, x_i) = 1$ and $d(x_i, x_j) = 2$ for $i \neq j$. It will be convenient to give explicit coordinates vectors for the $x_1, x_2, x_3, \mu$ embeddings. Hence, for reference, we begin by labeling $x^E_1, x^E_2, x^E_3, \mu^E$ as the Euclidean embeddings given in Figure 1, for the sake of disambiguation:

$$\mu^E = \begin{bmatrix} 0 \\ 0 \end{bmatrix}, x^E_1 = \begin{bmatrix} 1 \\ 0 \end{bmatrix}, x^E_2 = \begin{bmatrix} -\frac{1}{2} \\ \frac{\sqrt{3}}{2} \end{bmatrix}, x^E_3 = \begin{bmatrix} -\frac{1}{2} \\ -\frac{\sqrt{3}}{2} \end{bmatrix}$$

We will now embed these points into $\mathbb{H}^2_K$ using the Riemannian exponential map at the origin of the hyperboloid, $\mu^H$, which has coordinate form:

$$\mu^H = \begin{bmatrix} \frac{1}{\sqrt{|K|}} \\ 0 \\ 0 \end{bmatrix}$$

We must prepend a 0 to the Euclidean vectors prior to using them with the above exponential map formulation in order to properly treat them as tangent vectors (vectors lying in the tangent space of $\mathbb{H}^2_K$ with point of tangency $\mu^H$). We abuse notation slightly and treat the original Euclidean vectors equivalently to those with this augmentation. Computing the embedding, we see:

$$\exp^K_{\mu^H}(\mu^E) = \cosh(\sqrt{|K|} \cdot 0) \mu^H + 0 = \mu^H$$

$$x^H_1 := \exp^K_{\mu^H}(x^E_1) = \exp^K_{\mu^H}\left(\begin{bmatrix} 0 \\ 1 \\ 0 \end{bmatrix}\right) = \cosh(\sqrt{|K|}) \mu^H + \frac{\sinh(\sqrt{|K|})}{\sqrt{|K|}} \begin{bmatrix} 0 \\ 1 \\ 0 \end{bmatrix} = \begin{bmatrix} \frac{\cosh(\sqrt{|K|})}{\sqrt{|K|}} \\ \frac{\sinh(\sqrt{|K|})}{\sqrt{|K|}} \\ 0 \end{bmatrix}$$

$$x^H_2 := \exp^K_{\mu^H}(x^E_2) = \exp^K_{\mu^H}\left(\begin{bmatrix} 0 \\ -\frac{1}{2} \\ \frac{\sqrt{3}}{2} \end{bmatrix}\right) = \cosh(\sqrt{|K|}) \mu^H + \frac{\sinh(\sqrt{|K|})}{\sqrt{|K|}} \begin{bmatrix} 0 \\ -\frac{1}{2} \\ \frac{\sqrt{3}}{2} \end{bmatrix} = \begin{bmatrix} \frac{\cosh(\sqrt{|K|})}{\sqrt{|K|}} \\ \frac{-\sinh(\sqrt{|K|})}{2\sqrt{|K|}} \\ \frac{\sqrt{3}\sinh(\sqrt{|K|})}{2\sqrt{|K|}} \end{bmatrix}$$

$$x_3^H := \exp_{\mu^H}^K(x_3^E) = \exp_{\mu^H}^K \left( \begin{bmatrix} 0 \\ -\frac{1}{2} \\ -\frac{\sqrt{3}}{2} \end{bmatrix} \right) = \cosh(\sqrt{|K|})\mu^H + \frac{\sinh(\sqrt{|K|})}{\sqrt{|K|}} \begin{bmatrix} 0 \\ -\frac{1}{2} \\ -\frac{\sqrt{3}}{2} \end{bmatrix} = \begin{bmatrix} \frac{\cosh(\sqrt{|K|})}{\sqrt{|K|}} \\ \frac{-\sinh(\sqrt{|K|})}{2\sqrt{|K|}} \\ \frac{-\sqrt{3}\sinh(\sqrt{|K|})}{2\sqrt{|K|}} \end{bmatrix}$$

Notice that because the Riemannian exp is a local isometry, we have $d_{\mathbb{H}}^K(\mu^H, x_i^H) = 1$, preserving the original Euclidean distances to the origin in the tangent space. Indeed, we can confirm for $\mu^H$ and $x_1^H$:

$$d_{\mathbb{H}}^K(\mu^H, x_1^H) = d_{\mathbb{H}}^K \left( \begin{bmatrix} \frac{1}{\sqrt{|K|}} \\ 0 \\ 0 \end{bmatrix}, \begin{bmatrix} \frac{\cosh(\sqrt{|K|})}{\sqrt{|K|}} \\ \frac{\sinh(\sqrt{|K|})}{\sqrt{|K|}} \\ 0 \end{bmatrix} \right) = \frac{1}{\sqrt{|K|}} \cosh^{-1} \left( K \cdot -\frac{\cosh(\sqrt{|K|})}{|K|} \right) = 1$$

Thus we preserve the distances from $\mu$ to the children with no distortion. Now for the leaf node distortion analysis, we select $x_1^H$ and $x_2^H$, without loss of generality (by symmetry). Note:

$$d_{\mathbb{H}}^K(x_1^H, x_2^H) = d_{\mathbb{H}}^K \left( \begin{bmatrix} \frac{\cosh(\sqrt{|K|})}{\sqrt{|K|}} \\ \frac{\sinh(\sqrt{|K|})}{\sqrt{|K|}} \\ 0 \end{bmatrix}, \begin{bmatrix} \frac{\cosh(\sqrt{|K|})}{\sqrt{|K|}} \\ \frac{-\sinh(\sqrt{|K|})}{2\sqrt{|K|}} \\ \frac{\sqrt{3}\sinh(\sqrt{|K|})}{2\sqrt{|K|}} \end{bmatrix} \right)$$

$$= \frac{1}{\sqrt{|K|}} \cosh^{-1} \left( -\frac{K\cosh^2(\sqrt{|K|})}{|K|} - \frac{K\sinh^2(\sqrt{|K|})}{2|K|} \right)$$

$$= \frac{1}{\sqrt{|K|}} \left( \cosh^2(\sqrt{|K|}) + \frac{1}{2}\sinh^2(\sqrt{|K|}) \right)$$

Now observe:

$$\lim_{|K|\to 0^+} d_{\mathbb{H}}^K(x_1^H, x_2^H) = \lim_{|K|\to 0^+} \frac{1}{\sqrt{|K|}} \left( \cosh^2(\sqrt{|K|}) + \frac{1}{2}\sinh^2(\sqrt{|K|}) \right) = \sqrt{3}$$

and

$$\lim_{|K|\to\infty} d_{\mathbb{H}}^K(x_1^H, x_2^H) = \lim_{|K|\to\infty} \frac{1}{\sqrt{|K|}} \left( \cosh^2(\sqrt{|K|}) + \frac{1}{2}\sinh^2(\sqrt{|K|}) \right) = 2$$

Thus we see when the curvature is 0, we obtain the distorted Euclidean distance $\sqrt{3}$, as anticipated. Yet as the curvature becomes more and more negative, $d_{\mathbb{H}}^K(x_1^H, x_2^H)$ approaches 2, which is the original graph distance between any two leaf nodes. Thus we have manifested a hyperbolic embedding and have demonstrated that it is suitable to preserve the original graph distances in Figure 1 with arbitrarily low distortion.

## B Hyperbolic Geometry: Gromov $\delta$-Hyperbolicity Intuition

In this section, we follow up on the algebraic definition of Gromov $\delta$-hyperbolicity given in Section 3.4 and aim to give a more intuitive way of reasoning about this commonly used graph characterization metric. In service of attaining this end, we give a less algebraic and more geometrically intuitive definition of Gromov

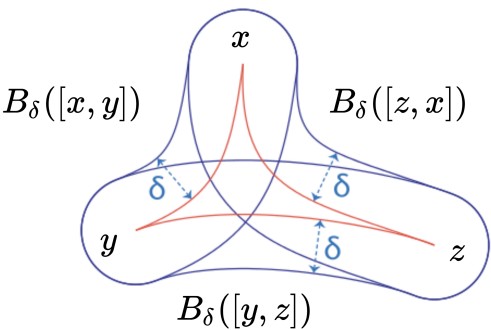

Figure 3: An illustration of the $\delta$-slim triangle condition for the metric space $(X, d)$. Recall $B_\delta(S) :=$ $\{p|\inf_{s \in S} d(s, p) < \delta\}$.

$\delta$-hyperbolicity in what follows.[5] Let $(X, d)$ be a geodesic metric space (a space in which any two points can be connected via a geodesic, not necessarily unique). Let $x, y, z \in X$. A geodesic triangle with vertices $x, y, z$ is the union of the three geodesic segments $[x, y], [y, z], [z, x]$ (where $[p, q]$ denotes a geodesic segment with endpoints $p$ and $q$). If for any point $m \in [x, y]$ there is a point in $[y, z] \cup [z, x]$ at distance less than $\delta$ of $m$, and similarly for points on the other edges, and $\delta \geq 0$, then the triangle is said to be $\delta$-slim. Please see Figure 3 for an illustration of this condition. We can then define a $\delta$-hyperbolic space as a geodesic metric space in which all geodesic triangles are $\delta$-slim. Notably, if $\delta = 0$, we see the triangles collapse to tripods, implying the space is isomorphic to a tree.

## C   Graph Characterization via Ollivier-Ricci Curvature

Though Gromov $\delta$-hyperbolicity is commonly used to characterize the geometry of graphs, we note that it assigns a single integer to an entire graph, and fails to distinguish finer graph geometry that may be relevant in a machine learning context. In particular, two trees with very different structure (e.g. very different branch factor) will both be assigned a Gromov $\delta$-hyperbolicity value of 0. Thus, one may desire a finer way to characterize graph geometry in this context. One such way to obtain a more fine characterization is via Ollivier-Ricci curvature (Lin et al., 2011). Ollivier-Ricci curvature is a discretization of Ricci curvature, and can be used to locally characterize graph edges. The definition is given below.

**Definition 1 (Ollivier-Ricci Curvature).**   Let $(X, d)$ be a separable, completely metrizable, metric space. Let $(m_x)_{x \in X}$ be a family of probability measures on $X$ such that (i) the measure $m_x$ depends measurably on $x \in X$ and (ii) for every $x \in X$, the first moment $\int d(x, y) \, dm_x(y)$ is finite. Let $x, y \in X, x \neq y$. The Ollivier-Ricci curvature $\kappa(x, y)$ of $(X, d, (m_x))$ along $(xy)$ is defined by:

$$\kappa(x, y) := 1 - \frac{\mathcal{T}_1(m_x, m_y)}{d(x, y)}$$

where $\mathcal{T}_1(m_x, m_y)$ is the $L^1$ transportation distance from $m_x$ to $m_y$.

In the context of graphs, we measure the Ollivier-Ricci curvature of an edge $(x, y)$ by taking $m_x$ and $m_y$ to be discrete uniform distributions over the neighbor sets $\mathcal{N}_x, \mathcal{N}_y$, respectively. The metric $d$ is induced by the graph (i.e. shortest path). Computing the curvature boils down to computing the $L^1$ transport distance between these distributions, which boils down to solving a linear program (and can be automated with a simple linear program solver). Computing the Ollivier-Ricci curvature for all edges can give a nuanced view of graph geometry that is not offered by Gromov $\delta$-hyperbolicity. In particular, note that while the graphs from Disease (Chami et al., 2019) and Tree1111 (a dataset we introduce) are both trees, i.e. Gromov $\delta$-hyperbolicity is 0 for both, the Ollivier-Ricci curvature profiles given in Figure 4 differ considerably. As such, Ollivier-Ricci curvature provides a more granular approach to graph characterization that would benefit

---

[5]This definition of $\delta$-hyperbolicity is equivalent to the prior definition up to a constant factor (Aksoy & Jin, 2013).

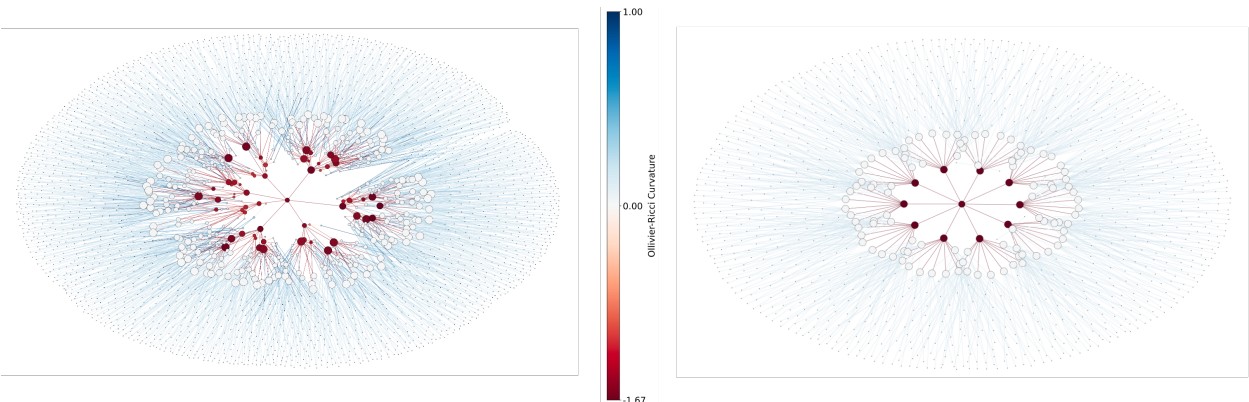

Figure 4: Ollivier-Ricci curvature (Lin et al., 2011) for two graphs. The left graph is the Disease (Chami et al., 2019) dataset and the right graph is a tree with a branch factor of 10 from the synthetic graph dataset Tree1111 we introduce.

the hyperbolic graph machine learning community in particular (this community currently largely relies on Gromov $\delta$-hyperbolicity).

## D   Tuning Euclidean Baselines

In cases where Euclidean baselines are not well presented, most of the time Euclidean baselines are not severely buggy, but rather, are not tuned to maximally present the extent of their performance. In this section, we investigate baselines in the application of geometric machine learning to the knowledge graph tasks in Chami et al. (2020). In particular, we note that further tuning the Euclidean models for one of the datasets that "exhibits hierarchical structures" (Chami et al., 2020) considerably bridges the gap between the baseline Euclidean models and the introduced hyperbolic models. Results are given in Table 3 (best of 5 trials is given; only the best result is reported, following conventions set by prior work for these tasks).

| | **Dataset** | WN18RR (Bordes et al., 2013) | | | |
|---|---|---|---|---|---|
| Space | Model | MRR | H@1 | H@3 | H@10 |
| $\mathbb{R}^d$ | RefE | 0.455 | 0.419 | 0.470 | 0.521 |
| | RotE | 0.463 | 0.426 | 0.477 | 0.529 |
| $\mathbb{R}^d$ | RefE (tuned) | 0.457 | 0.425 | 0.469 | 0.512 |
| | RotE (tuned) | 0.466 | **0.429** | 0.481 | 0.534 |
| $\mathbb{H}^d$ | RefH | 0.447 | 0.408 | 0.464 | 0.518 |
| | RotH | **0.472** | 0.428 | **0.490** | **0.553** |

Table 3: Above we give link prediction results for WN18RR (Bordes et al., 2013), a dataset that "exhibits hierarchical structures" (Chami et al., 2020). We deal with low-dimensional embeddings ($d = 32$) in the filtered setting, as was the case in the original paper (Chami et al., 2020). The properly tuned Euclidean models considerably close the gap between the baseline Euclidean results and the hyperbolic results. While tuning, we focus on MRR, but most other metrics also exhibit a concomitant increase.

## E   Analysis of Graph Methods when Features are Uninformative

In this section, under some assumptions, we analyze a variety of graph machine learning methods in a context where the features are uninformative of the graph structure. We do this to better understand how the graph is being used when the features offer no distinction between nodes for the task at hand.

**Per-layer Analysis**

We analyze a variety of common graph machine learning baselines on a graph dataset where the features are i.i.d.; this is a very strong setting, in that when this is the case, graph methods must use the graph to obtain nontrivial performance. Note that our introduced synthetic Tree1111 ($\gamma = 0$) dataset falls under this setting. Recall that a graph dataset is given by the graph structure $G = (V, E)$, whence the adjacency matrix $A$ is derived, together with the associated feature vectors $X \in \mathbb{R}^{|V| \times n}$; that is, a feature vector in $\mathbb{R}^n$ is given for each of the vertices. We will make the simplifying assumption that the input features are drawn i.i.d. from $\mathcal{N}(\mathbf{0}, I_d)$.

**Multi-layer Perceptron (MLP)** Recall that a layer of the MLP is simply:

$$X^{(n)} = \sigma(X^{(n-1)}W)$$

where $W$ are the learned weights, $X^{(n-1)}$ is the matrix of input features, and $\sigma$ is a non-linearity. This is one of the simplest models used in this context and is a baseline that uses no graph information (only the features).

**Graph Convolutional Network (GCN)** A layer of the GCN (Kipf & Welling, 2017) is given by:

$$X^{(n)} = \sigma(\tilde{A}X^{(n-1)}W)$$

where $W$ and $X^{(n-1)}$ are as given above, and $\tilde{A}$ is the normalized adjacency matrix of the underlying graph. We have:

$$\tilde{A} = \bar{D}^{-1/2}\bar{A}\bar{D}^{-1/2}$$

$$\bar{A} = A + I_{|V|}$$

where $\bar{D}$ is the degree matrix of $\bar{A}$. The GCN is the simplest modern graph machine learning method that uses graph information by way of a normalized adjacency matrix multiplication.

**Graph Attention Network (GAT)** A layer of GAT (Velickovic et al., 2018) is given by:

$$X_i^{(n)} = \sigma\left(\sum_{j \in \mathcal{N}_i} \alpha_{ij} W X_j^{(n-1)}\right)$$

where $\mathcal{N}_i$ represents the graph neighbors of node $i$ and:

$$\alpha_{ij} = \frac{\exp\left(\text{LeakyReLU}\left(a^T \begin{bmatrix} WX_i^{(n-1)} \\ WX_j^{(n-1)} \end{bmatrix}\right)\right)}{\sum_{k \in \mathcal{N}_i} \exp\left(\text{LeakyReLU}\left(a^T \begin{bmatrix} WX_i^{(n-1)} \\ WX_k^{(n-1)} \end{bmatrix}\right)\right)}$$

are the attention weights. Note that if all node features $X_i$ are i.i.d. Gaussian, the vectors $\begin{bmatrix} WX_i^{(n-1)} \\ WX_j^{(n-1)} \end{bmatrix}$ are all equal in distribution for any $i, j$. By properties of softmax, this implies $\alpha_{ij}$ concentrates on $\frac{1}{|\mathcal{N}_i|}$, and hence $X_i^{(n)}$ concentrates on $\sigma\left(\sum_{j \in \mathcal{N}_i} \frac{1}{|\mathcal{N}_i|} W X_j^{(n-1)}\right)$, and since $W$ is constant, the features over which learning happens approximately follow $\mathcal{N}(\mathbf{0}, \frac{1}{|\mathcal{N}_i|}I_d)$.

That is to say, the attention scores are spread evenly across all neighbors, and use of the graph structure reduces simply to scaling down covariance by degree.

**Simple Graph Convolution Network (SGC)** A K-layer SGC (Wu et al., 2019) is given by:

$$X^{(n)} = \text{softmax}(\tilde{A}^K XW)$$

where $\tilde{A}$ is the normalized adjacency matrix. On a per-layer basis ($K = 1$) it reduces to:

$$X^{(n)} = \text{softmax}(\tilde{A}XW)$$

that is, the graph is used in no more complicated of a way than how it is used for the GCN.

**Graph Sample and Aggregation (SAGE)** A layer of SAGE (Hamilton et al., 2017) is given by:

$$X_i^{(n)} = \sigma \left( W \begin{bmatrix} X_i^{(n-1)} \\ X_{\mathcal{N}(i)}^{(n)} \end{bmatrix} \right)$$

where:

$$X_{\mathcal{N}(i)}^{(n)} = \text{Agg}^{(n)}(\{X_j^{(n-1)}, \forall j \in \mathcal{N}(i)\})$$

The aggregation function Agg is selected to be either a mean or max pool operator. For the sake of simplicity, let us assume the aggregation operator is the mean. Note if the aggregation is the mean function, we have:

$$X_{\mathcal{N}(i)}^{(n)} \stackrel{d}{=} \mathcal{N}(\mathbf{0}, \frac{1}{|\mathcal{N}_i|}I_d)$$

That is to say, $X_i^{(n)}$ is learned from the previous feature vector and a feature drawn from a distribution with covariance scaled down by the degree. In other words, merely the degree is used from the graph and the model uses graph structure in no more a sophisticated manner than the GCN.

A summary of this analysis is provided in Table 4, highlighting how each method does (or does not) use the graph. As we can clearly see, when dealing with a dataset where the features are not informative, most graph learning methods do not actually use the graph information in a more sophisticated way than the GCN. This may perhaps be surprising and showcases the overreliance on features of most graph methods. If graph information is truly necessary to solve the task at hand, this analysis indicates that coming up with positional embeddings may prove to be highly beneficial (and this is confirmed in Figure 2 of the main paper).

Table 4: Summary Analysis of Various Graph Machine Learning Methods on a Dataset with Independent, Identically Distributed Gaussian Features

| Method | Layer Structure | Graph Dependence Reduction (first layer at initialization) |
|---|---|---|
| MLP | $X^{(n)} = \sigma(X^{(n-1)}W)$ | None |
| GCN | $X^{(n)} = \sigma(\tilde{A}X^{(n-1)}W)$ | $\tilde{A}X^{(n-1)}W$ |
| GAT | $X_i^{(n)} = \sigma\left(\sum_{j\in\mathcal{N}_i}\alpha_{ij}WX_j^{(n-1)}\right)$, where $\alpha_{ij} = \dfrac{\exp\left(\text{LeakyReLU}\left(a^T\begin{bmatrix}WX_i^{(n-1)}\\WX_j^{(n-1)}\end{bmatrix}\right)\right)}{\sum_{k\in\mathcal{N}_i}\exp\left(\text{LeakyReLU}\left(a^T\begin{bmatrix}WX_i^{(n-1)}\\WX_k^{(n-1)}\end{bmatrix}\right)\right)}$ | $\mathcal{N}(\mathbf{0}, \frac{1}{|\mathcal{N}_i|}I_d)$ |
| SGC | $X^{(n)} = \text{softmax}(\tilde{A}^K X^{(n-1)}W)$ | $\tilde{A}X^{(n-1)}W$ |
| SAGE | $X_i^{(n)} = \sigma\left(W\begin{bmatrix}X_i^{(n-1)}\\X_{\mathcal{N}(i)}^{(n)}\end{bmatrix}\right)$, where $X_{\mathcal{N}(i)}^{(n)} = \text{Agg}^{(n)}(\{X_j^{(n-1)}, \forall j \in \mathcal{N}(i)\})$ | $\mathcal{N}(\mathbf{0}, \frac{1}{|\mathcal{N}_i|}I_d)$, if $\text{Agg}^{(n)}$ is mean |

## F HGCN (Chami et al., 2019) Euclidean Multi-layer Perceptron Bug

A fairly prominent, and motivating, discovery in our paper is that in Chami et al. (2019), there was a bug present for Euclidean baselines, most prominently for the Euclidean MLP. This bug is manifested by a single line of code in the publicly released implementation; this line constrained Euclidean model representations to lie within the unit ball, perhaps a vestigial feature from the Poincaré ball implementation. For reference, the line, together with the relevant conditional statement, is given below (lines 104-105 of 'models/base_models.py' of the original repository, accessible at https://github.com/HazyResearch/hgcn):

```python
if self.manifold_name == 'Euclidean':
    h = self.manifold.normalize(h)
```

If you simply comment these lines, as we did above for Table 1, and slightly modify the Fermi-Dirac decoder in 'layers/layers.py' by replacing line 86:

```python
probs = 1. / (torch.exp(((dist - self.r) / self.t)) + 1.0)
```

with:

```python
probs = 1. / (torch.exp(((dist - self.r) / self.t).clamp_max(20)) + 1.0)
```

for better numerical stability, the very Euclidean MLP presented in the original GitHub repository[6] attains 98.7% test ROC AUC for link prediction on Disease (up from 72.6% for the original version) and 80.3% test F1 score for node classification on Disease (up from 28.8% for the original version). The Euclidean MLP also attains 99.1% test ROC AUC for link prediction on Disease-M (up from 55.3%). Note that Disease and Disease-M (Chami et al., 2019) are the most hyperbolic datasets presented, as measured by Gromov $\delta$-hyperbolicity (the $\delta$-hyperbolicity is 0, meaning the graphs from both datasets are trees). This is a considerable issue, since the HGCN GitHub repository has been used in a number of hyperbolic machine learning papers (Chen et al., 2021; Lou et al., 2020; Katsman et al., 2023) that introduce their own hyperbolic models.

## G   Experimental Details

Our runs for Table 1, specifically the "MLP (debugged)" row, were performed on a single NVIDIA GeForce RTX 3090 GPU. All reported results in Table 1 give mean and standard deviation over 5 trials; trials differ only in terms of the seed used. The same procedure is followed for the three rows of Table 2 and each data point of Figure 2. Data for other rows in Table 1 was taken from existing results reported in Chami et al. (2019) and Chen et al. (2021).

A complete repository with our code (and a bug-free version of the Euclidean multi-layer perceptron model from Chami et al. (2019)) is available at the following Github link. There, our README gives explicit line-by-line commands to reproduce our results. The hyperparameters for our best runs were found with tuning via the Weights and Biases Biewald (2020) framework and we include the YAML configuration file we used for tuning in the repository as well.

---

[6]Please find the original GitHub repository for Chami et al. (2019) at https://github.com/HazyResearch/hgcn.

