# OpenReview forum: "Shedding Light on Problems with Hyperbolic Graph Learning"
_TMLR — Accepted by TMLR_

### Review · Reviewer_iz4Q · 2024-11-25

**Summary Of Contributions:**

The authors demonstrate a clear issue in, at the very least, some related work on hyperbolic graph learning. Specifically, they argue and show that (1) euclidean baselines were not fairly represented, resulting in misleading conclusions (2) some architectural design choices in hyperbolic graph learning models are not supported by the underlying theory (3) a standard measure of hybolic relevance, the Gromov \delta-hyperbolicity, is not a good measure for suitedness of a dataset for evaluation of hyperbolic graph learning models.

**Audience:**

Yes

**Broader Impact Concerns:**

Since the authors propose synthetic tree datasets to replace more practically relevant datasets currently in use, the authors could clarify that this proposal could lead to a different form of spurious evaluation of only synthetic benchmarks, and practically relevant datasets are sorely needed. While the authors include this to some extent in their "Future Work" paragraph, assuming this shortcoming is not resolved, they should include this more explicitly in their "Impact statement" paragraph.

**Claims And Evidence:**

Yes

**Requested Changes:**

Each weakness above is essentially also a requested change:
- weaken the aggressive language in particular directed towards Chami et al. (2019) (critical)
- Include a paragraph in the introduction explaining criticism of Chami et al. (2019) and other works. (non-critical)
- Weaken their strong language regarding Gromov \delta-hyperbolocity. (critical)
- Put Ollivier-Ricci curvature and its comparison to Gromov \delta-hyperbolicity into proper context. (critical)
- Clarify the proposal of GNN baselines as a solution to the mentioned problems (critical)
- Clarify the proposal of synthetic tree datasets like Tree111 as a solution to the mentioned problems (critical)
- Add an experimental setup for hyperparameter tuning of Euclidean baselines. (critical)
- Add statistical significance evaluations (non-critical)
- Fix minor typo in appendix (non-critical)

**Strengths And Weaknesses:**

Strengths:
- The authors very convincingly demonstrate their first point, i.e., that Euclidean baselines were not fairly represented previously.
- The authors do a good job of arguing the shortcomings of previous work.
- The authors focus on fair evaluations given this particular state of affairs.
- The authors also attempt to explain how these issues might have arisen.

Weaknesses:
- Large parts of this work are antithetical to currently held beliefs on model evaluation, which is not an issue per se. The authors try to defuse their, for lack of a better word, aggressive language towards some of the related works. However, some parts of this work could still be considered personal attacks. In particular, there is a large focus on Chami et al. (2019) (and then Chami et al. (2020)). I would suggest that the authors, wherever possible, always include all related works that made these mistakes to, at the very least, diffuse the blame among all of these. Namely, Chen et al. (2021), Zhang et al. (2019), and Zhang et al. (2021), and others.
- (in relation to the previous weakness) I would also suggest. However, I will not make my score dependent on this that the authors add a paragraph to their introduction, putting Chami et al. (2019) and the other works into context, in the sense that the authors are not attempting to attack these other authors but are only criticizing some aspects of their work. If possible, this could also include a recognition of work well done in these related works, highlighting that there are only some issues in evaluation and model design, as mentioned.
- In section 4.2, "Poor geometric characterization of graph datasets," the authors claim that Gromov \delta-hyperbolicity is a poor characterization, especially due to its coarseness in treating trees. While I acknowledge that this might be the case, one mentioned issue, i.e., that node features are not taken into account, is a fundamental issue in many areas of graph learning and, to the best of my knowledge, has no current solution. By this, I mean that this cannot be used as an argument that Gromov \delta-hyperbolicity is a poor measure. By this standard, almost every measure is a poor measure. I suggest that the authors tone down their language in this paragraph and put into context that this is an important area of study for which we currently do not have a solution.
- (in relation to the previous weakness) Along the same lines, the authors claim that Ollivier-Ricci curvature is a better measure since it is more fine-grained. The authors do not demonstrate that coarseness/fine-grainedness is particularly relevant or beneficial; however, they do not demonstrate that Ollivier-Ricci curvature is a better measure. I suggest that the authors either add further arguments, carefully reconsider the strength and tone of their statements, or remove this paragraph (which would necessitate other changes).
- In section 5, "Euclidean MLP baseline," The authors claim that after a Euclidean MLP fails to perform well, a GNN should act as an additional baseline. This in and of itself makes sense, of course. However, the authors should more clearly articulate that no author can guarantee that this GNN baseline benchmark will be a fair comparison since it has been demonstrated repeatedly that different GNNs perform vastly differently depending on the benchmarking dataset in question. Perhaps the most striking recent publication to this effect would be Bechler-Speicher et al. (2024).
- In section 5, "Defining Tree(...)" and "Tree1111 dataset," the authors claim that their notably strongly synthetic dataset is a better benchmark for hyperbolic graph learning than Disease and Disease-M, which are also synthetic but strongly motivated, representing real processes. This is a poor argument as the authors compare their purely academically relevant synthetic dataset to datasets that might have practical relevance. I suggest that the authors either provide a better-suited practical dataset (even some dataset that is already in use) or make it clear that they do not have a better-suited practical dataset and that they thus propose a synthetic replacement that is at least better suited in the sense that graph structure becomes relevant. To be more specific, I propose that the authors tone down their language and make it clear to any prospective reader that their proposal is also not perfect when it is being proposed.
- The authors do not provide an experimental setup or details on their experimental setup regarding their tuning on the Euclidean MLP models. I am assuming that this tuning includes the tuning of model hyperparameters, in which case the authors should clearly state how these hyperparameters are being tuned.
- The authors criticize previous work for not achieving statistically significant results but don't clarify whether their tables (notably Table 1) include significance tests. While I believe it is quite apparent that all presented results are statistically significant, these tests should be explicitly conducted and included in this work. Specifically, the author should test whether each second-best method (the best being the bold entry) performs statistically significantly worse than the best method and mark each second (third and so on) method that does not perform significantly worse by underlining it or making this clear in some other way. This also necessitates an additional explanation in the table's caption.
- There is a small mistake in Appendix A "d(\mu, x_i) = 1 and d(x_i, x_i)=2 for ...". It should be "d(x_i, x_j)=2".

Maya Bechler-Speicher et al., "Graph Neural Networks Use Graphs When They Shouldn't", ICML (2024)

---

> ### Author Response · Authors · 2024-12-23
> **Author Response (1 of 3)**
>
> Thank you for the review and the constructive comments. We appreciate that you note our paper “convincingly demonstrates” that “Euclidean baselines were not fairly represented previously.” We address your comments from the “Weaknesses” section below together with the parallel comments in the “Requested Changes” section.
>
> ---
>
> > Large parts of this work are antithetical to currently held beliefs on model evaluation, which is not an issue per se. The authors try to defuse their, for lack of a better word, aggressive language towards some of the related works. However, some parts of this work could still be considered personal attacks. In particular, there is a large focus on Chami et al. (2019) (and then Chami et al. (2020)). I would suggest that the authors, wherever possible, always include all related works that made these mistakes to, at the very least, diffuse the blame among all of these. Namely, Chen et al. (2021), Zhang et al. (2019), and Zhang et al. (2021), and others.
>
> A: This point is well taken. Certainly, what we are highlighting here is pertinent to a series of papers (not just Chami et al. (2019)) and we aim to be explicitly conciliatory in that we seek concrete solutions that will improve evaluation and model design in this field. We have decreased the number of references to Chami et al. and have changed several places to diffuse responsibility across the subfield.
>
> ---
>
> > (in relation to the previous weakness) I would also suggest. However, I will not make my score dependent on this that the authors add a paragraph to their introduction, putting Chami et al. (2019) and the other works into context, in the sense that the authors are not attempting to attack these other authors but are only criticizing some aspects of their work. If possible, this could also include a recognition of work well done in these related works, highlighting that there are only some issues in evaluation and model design, as mentioned.
>
> A: Certainly; these cited papers had a number of good ideas and were very influential. We will include a paragraph in the updated version of our paper that will note the interesting and novel aspects of the contributions of the aforementioned previous work.
>
> ---
>
> > In section 4.2, "Poor geometric characterization of graph datasets," the authors claim that Gromov \delta-hyperbolicity is a poor characterization, especially due to its coarseness in treating trees. While I acknowledge that this might be the case, one mentioned issue, i.e., that node features are not taken into account, is a fundamental issue in many areas of graph learning and, to the best of my knowledge, has no current solution. By this, I mean that this cannot be used as an argument that Gromov \delta-hyperbolicity is a poor measure. By this standard, almost every measure is a poor measure. I suggest that the authors tone down their language in this paragraph and put into context that this is an important area of study for which we currently do not have a solution.
>
> A: We have augmented our current writing to emphasize this point. Certainly, even the suggested Ollivier-Ricci curvature does not account for node features and this is a point well worth making. Our intention was primarily to push the field to come up with ways to better characterize graph datasets, both the graph and the node features. There are currently tools to better characterize the graph, but tools to better characterize node features (alone and with respect to the graph) are essentially non-existent; we thus believe this would make for a fruitful line of research for future work.
>
> ---
>
> > (in relation to the previous weakness) Along the same lines, the authors claim that Ollivier-Ricci curvature is a better measure since it is more fine-grained. The authors do not demonstrate that coarseness/fine-grainedness is particularly relevant or beneficial...
>
> A: Our primary purpose in mentioning Ollivier-Ricci curvature was to show that there are more nuanced and rich characterizations of graph geometry, and moreover, that there characterizations offer greater separation between structures that may be viewed as nearly identical by other characterizations (e.g. Gromov $\delta$\-hyperbolicity). Ideally, one could use the curvature profile of a graph to come up with a bespoke embedding space best suited for the graph; in fact, a coarse version of this idea has been attempted by Gu et al. (https://openreview.net/pdf?id=HJxeWnCcF7), but the precise mechanics of such an idea are quite complicated since local geometry is hard to capture with a simple product space. You are also correct about the fact that this does not solve the problem of analyzing the node features together with the graph; one would need to develop a more fundamental set of tools to perform this analysis and we believe this is a very fruitful avenue for future work.

---

> ### Author Response · Authors · 2024-12-23
> **Author Response (2 of 3)**
>
> > In section 5, "Euclidean MLP baseline," The authors claim that after a Euclidean MLP fails to perform well, a GNN should act as an additional baseline. This in and of itself makes sense, of course. However, the authors should more clearly articulate that no author can guarantee that this GNN baseline benchmark will be a fair comparison since it has been demonstrated repeatedly that different GNNs perform vastly differently depending on the benchmarking dataset in question. Perhaps the most striking recent publication to this effect would be Bechler-Speicher et al. (2024).
>
> A: Yes, you are certainly correct. We have cited Bechler-Speicher et al. (2024) and have added a note in the main text that there is no guarantee this will make for a better baseline since different GNNs perform differently and it may even be the case that the graph is not necessary (perhaps even detrimental) for a given task.
>
> ---
>
> > In section 5, "Defining Tree(...)" and "Tree1111 dataset," the authors claim that their notably strongly synthetic dataset is a better benchmark for hyperbolic graph learning than Disease and Disease-M, which are also synthetic but strongly motivated, representing real processes. This is a poor argument as the authors compare their purely academically relevant synthetic dataset to datasets that might have practical relevance. I suggest that the authors either provide a better-suited practical dataset (even some dataset that is already in use) or make it clear that they do not have a better-suited practical dataset and that they thus propose a synthetic replacement that is at least better suited in the sense that graph structure becomes relevant. To be more specific, I propose that the authors tone down their language and make it clear to any prospective reader that their proposal is also not perfect when it is being proposed.
>
> A: Certainly, we can make the fact that our synthetic dataset is not perfect more clear. It is quite synthetic, as are Disease and Disease-M, perhaps moreso, and we admit that it was constructed for a fairly particular purpose: i.e. for the purpose of illustrating that (1) a dataset with uninformative features would force use of the graph (to perform nontrivially) and that (2) introducing the graph structure in the features can obviate the need for the graph itself. These are two crucial points that are very underemphasized and underexplored in both the hyperbolic graph learning literature and more broadly in graph learning literature. This dataset suffices for these purposes, although we certainly admit that coming up with a number of augmentations that better demonstrate real-world phenomena is an interesting line for future work that exceeds our original scope. We believe that the design of interesting synthetic datasets to demonstrate certain important aspects of graph learning is an important avenue for future work.
>
> ---
>
> > The authors do not provide an experimental setup or details on their experimental setup regarding their tuning on the Euclidean MLP models. I am assuming that this tuning includes the tuning of model hyperparameters, in which case the authors should clearly state how these hyperparameters are being tuned.
>
> A: In our attached zip file (supplementary material), we included a thorough README with exact commands and hyperparameters to reproduce the results in the paper for all datasets. We will make this explicitly clear in the updated version of Appendix G, which describes the experimental details in our paper. We tuned all hyperparameters by performing Weights and Biases (wandb) sweeps. The explicit .yaml file with hyperparameter sweep parameters is now included in the zip file and mentioned in the README. We will also add this detail to Appendix G.

---

> ### Author Response · Authors · 2024-12-23
> **Author Response (3 of 3)**
>
> > The authors criticize previous work for not achieving statistically significant results but don't clarify whether their tables (notably Table 1) include significance tests. While I believe it is quite apparent that all presented results are statistically significant, these tests should be explicitly conducted and included in this work. Specifically, the author should test whether each second-best method (the best being the bold entry) performs statistically significantly worse than the best method and mark each second (third and so on) method that does not perform significantly worse by underlining it or making this clear in some other way. This also necessitates an additional explanation in the table's caption.
>
> A: We do not criticize previous work for not achieving statistically significant results, notably, we criticize the test tasks used and state that they were often easy enough to not yield statistically significant separation between the performance of Euclidean baselines and state-of-the-art hyperbolic models. This is an important distinction and is evident from the means and standard deviations given in Table 1 for e.g. the Disease LP result in which the MLP attains $98.7 \pm 0.2$ versus $98.4 \pm 0.4$ for G-RResNet Horo. Ideally, a well-chosen test task should showcase the best hyperbolic method as outperforming the most basic Euclidean baseline by a lot. We believe the means and standard deviations given in prior work are absolutely sufficient to assess statistical significance. That being said, following your recommendation we have performed paired t-tests to obtain explicit confirmation and now we highlight the best result only if the result gives a p-value < 0.01 after running a paired significance t-test against the next best result, and have mentioned this in the table’s caption.
>
> ---
>
> > There is a small mistake in Appendix A "d(\mu, x_i) = 1 and d(x_i, x_i)=2 for ...". It should be "d(x_i, x_j)=2".
>
> A: Thank you for pointing this out, it has been fixed.
>
> ---
>
> > Since the authors propose synthetic tree datasets to replace more practically relevant datasets currently in use, the authors could clarify that this proposal could lead to a different form of spurious evaluation of only synthetic benchmarks, and practically relevant datasets are sorely needed. While the authors include this to some extent in their "Future Work" paragraph, assuming this shortcoming is not resolved, they should include this more explicitly in their "Impact statement" paragraph.
>
> A: Certainly, we have augmented the impact statement to make it clear that we do not want to push the field to the extreme of only synthetic benchmarks, and that practical real world datasets are ideal for evaluation.

---

### Review · Reviewer_krJo · 2024-12-05

**Summary Of Contributions:**

This work identifies systematic issues in hyperbolic graph learning, including poorly tuned Euclidean baselines, flawed task designs, and misuse of Gromov $\delta$-hyperbolicity. The authors demonstrate that well-tuned Euclidean models often outperform or compete with hyperbolic ones on benchmarks previously used to demonstrate the superiority of hyperbolic models. They also propose synthetic benchmarks to evaluate hierarchical structures, and suggest Ollivier-Ricci curvature as a better alternative to $\delta$-hyperbolicity. The work challenges key assumptions and offers actionable insights for improving evaluation practices in the field.

**Audience:**

Yes

**Claims And Evidence:**

Yes

**Requested Changes:**

* **(R1) Expand Real-World Benchmark Analysis**: Include results for all six datasets from Chami et al. (2019) or explain why certain datasets were excluded.
* **(R2) Clarify the Bug**: Provide more details about the buggy code in the Euclidean baselines, including its possible original purpose (i.e., why was it possibly introduced) and the specific changes that occur when it is removed.
* **(R3) Synthetic Dataset Design**:
  * **(R3a)** Consider introducing nonlinear dependencies between the features (e.g., transform the parent features with sigmoid or ReLU, perhaps before/after applying a non-diagonal linear transformation) to better simulate characteristics of real-world data.
  * **(R3b)** Consider exploring additional graph topologies beyond trees, such as grids or random graphs, to further explore the impact of graph topology/geometry across Euclidean and hyperbolic graph representation learning models.
   * **(R3c)** Expand the dataset to appropriate tasks beyond link prediction, such as node classification or graph regression.
  * **(R3d)** Compare/contrast the underlying model against the SIR model for disease propagation from Chami et al. (2019).
* **(R4) Ollivier-Ricci Curvature**: Better contextualize ORC in the work.
* **(R5) Experimental Details**: Expand Section G in the Appendix with more details on the datasets, baselines, hyperparameters, etc.

**Strengths And Weaknesses:**

## Strengths

The work is very relevant for the graph and geometric representation learning communities. Some of the work's strengths are:

* **Identifying Systematic Issues**: The paper properly identifies widespread problematic hyperbolic graph learning practices and systemic issues and proposes first steps to mitigate the situation.
* **Novel Benchmark**: The paper proposes a novel, controlled, synthetic benchmark to evaluate hierarchical graph learning models.
* **Impactful and Practical Recommendations**: The work emphasizes the importance of reproducibility, proper task design, and theoretical grounding, which are essential for advancing geometric machine learning.

## Weaknesses

The work makes a valuable contribution by identifying systematic issues in hyperbolic graph representation learning and providing important and actionable insights. However, to emphasize the broader applicability of its findings, a few gaps in analysis and presentation need to be addressed. These will be straightforward to address and would significantly enhance the impact of the work.
* **(W1) Limited Real-World Evaluation**: Table 1 considers only three of the six datasets from Chami et al. (2019). Expanding the evaluation to at least all six would provide a more comprehensive evaluation
* **(W2) Unclear Explanation of the Identified Bug**: The discussion of the bug in Euclidean baselines lacks detail. The authors do not fully explain the possible original intent of the buggy line (i.e., why it was possibly present) or how its removal changes the behavior of the model. It is not clear what researchers need to be careful about when implementing Euclidean baselines or how such bugs could be systematically identified and avoided in future work.
* **(W3) Synthetic Dataset Simplifications**:
  * **(W3a)** The Tree dataset assumes linear dependencies between adjacent node features, which may not adequately reflect the characteristics of real-world data.
  * **(W3b)** The dataset focuses exclusively on tree dependencies, and does not provide any insights for non-tree graph topologies (e.g., grids or random graphs).
  * **(W3c)** The dataset focuses exclusively on the link prediction task.
  * **(W3d)** The authors do not compare/contrast the design and rationale of their Tree model against the SIR disease propagation model from Chami et al. (2019)
* **(W4) Ollivier-Ricci Curvature**: While the authors discuss its potential as a finer metric for characterizing graph data that would benefit from hyperbolic representation learning, they do not integrate it into their experiments or benchmarks, leaving its practical utility unexplored.

---

> ### Author Response · Authors · 2024-12-23
> **Author Response (1 of 2)**
>
> Thank you for the review and the constructive comments. We appreciate that you note our paper “identifies widespread problematic” practices and systemic issues and that our paper “proposes a novel, controlled, synthetic benchmark” to evaluate hierarchical graph learning models. We address your comments from the “Weaknesses” section below together with the parallel comments in the “Requested Changes” section.
>
> ---
>
> > (W1/R1) Limited Real-World Evaluation: Table 1 considers only three of the six datasets from Chami et al. (2019). Expanding the evaluation to at least all six would provide a more comprehensive evaluation
>
> A: The primary purpose of our paper was to demonstrate that datasets previously thought to be the most hyperbolic, as measured by Gromov $\delta$-hyperbolicity, frequently do not benefit from currently hyperbolic graph neural networks, contradicting a somewhat well-established orthodoxy. Note that the lower the Gromov $\delta$-hyperbolicity, the more tree-like a graph is, with a $\delta$ of $0$ corresponding to a perfect tree. We initially focused our attention on 4 of the 6 most hyperbolic datasets in Chami et al. (2019), each of which has a $\delta$ of $0$ or $1$: Disease, Disease-M, Human PPI, Airport. Of these 4 datasets, only 3 were readily available in the original repo (Human PPI was excluded). As such, we focused on these three datasets in our paper. The remaining two datasets have substantially increased Gromov $\delta$-hyperbolicity: Pubmed with $\delta =3.5$ and Cora with $\delta=11$. The original paper was clear in that it did not expect the hyperbolic models to perform much better than the most sophisticated Euclidean models for e.g. Cora. After tuning the MLP on Cora for NC, we found that the original result can be improved to $69.4 \pm 1.3$ (over 5 trials). Although this is better than the original result of $51.5 \pm 1.0$ given in the paper, it is still considerably worse than the best Euclidean GCN result given by GAT (of $83.0 \pm 0.7$). Although this is a result we can include in the appendix for completeness, our primary focus is on the datasets previously claimed to be most hyperbolic (since this is what drives our narrative and investigation with respect to, among other things, differences between the graph and node features), hence our focus on Disease, Disease-M, and Airport.
>
> ---
>
> > (W2/R2) Unclear Explanation of the Identified Bug: The discussion of the bug in Euclidean baselines lacks detail. The authors do not fully explain the possible original intent of the buggy line (i.e., why it was possibly present) or how its removal changes the behavior of the model. It is not clear what researchers need to be careful about when implementing Euclidean baselines or how such bugs could be systematically identified and avoided in future work.
>
> A: We dedicate an appendix section (Appendix F) to discussing this bug, but we appreciate the suggestion and have added some of these details to the main paper. The central issue is that two lines in the implementation of the MLP normalize all Euclidean features to lie within the unit ball; commenting these two lines alleviates this issue. Although this introduces some numerical instability to the Fermi-Dirac decoder, there is also a simple one line fix for this that we also give in Appendix F. The removal of this bug allows for the Euclidean model/Euclidean representation space to be fully utilized, i.e., the features are no longer constrained to a unit ball. We do not believe we can accurately speculate on the original intent behind this change and/or why this bug was originally present in the code, but perhaps this was a vestigial feature from the Poincaré ball implementation, which has a unit ball representation space (although the metric is completely different, i.e., it is the hyperbolic metric).
>
> ---
>
> > (W3/R3) Synthetic Dataset Simplifications
>
> A: We admit that our synthetic dataset is fairly straightforward, but it was made primarily for the purpose of illustrating that (1) a dataset with uninformative features would force use of the graph (to perform nontrivially) and that (2) introducing the graph structure in the features can obviate the need for the graph itself. These are two crucial points that are very underemphasized and underexplored in both the hyperbolic graph learning literature and more broadly in graph learning literature. This dataset suffices for these purposes, although we certainly admit that a number of augmentations can make for a more rich collection of synthetic datasets that have uses exceeding our original scope. We believe that the design of interesting synthetic datasets to demonstrate certain important aspects of graph learning is an important avenue for future work.

---

> ### Author Response · Authors · 2024-12-23
> **Author Response (2 of 2)**
>
> > (W4/R4) Ollivier-Ricci Curvature: While the authors discuss its potential as a finer metric for characterizing graph data that would benefit from hyperbolic representation learning, they do not integrate it into their experiments or benchmarks, leaving its practical utility unexplored.
>
> A: Our primary purpose in mentioning Ollivier-Ricci curvature was to show that there are more nuanced and rich characterizations of graph geometry than Gromov $\delta$-hyperbolicity, and moreover, that theses characterizations offer greater separation between structures that may be viewed as nearly identical by other characterizations (e.g. Gromov $\delta$\-hyperbolicity). Ideally, one could use the Ollivier-Ricci curvature profile of a graph to come up with a bespoke embedding space best suited for the graph; in fact, a coarse version of this idea has been attempted by Gu et al. (https://openreview.net/pdf?id=HJxeWnCcF7), but the precise mechanics of such an idea are quite complicated since local geometry is hard to capture with a simple product space. We believe that before exploring this more practically, one has to address a more fundamental issue: we do not currently have a good way of characterizing the geometry of the node features together with the geometry of the graph. The Ollivier-Ricci curvature does not account for node features and this is a point well worth making. Despite the fact that this is arguably a better tool for analyzing the geometry of graphs than is currently being widely used, this means Ollivier-Ricci curvature does not provide a complete solution. One would need to develop a more fundamental set of tools to properly characterize node feature geometry together with the geometry of the graph; we believe this is a very fruitful and important avenue for future work following our paper. We will add this as an explicit clarification to the updated version of the paper.
>
> ---
>
> > (R5) Experimental Details: Expand Section G in the Appendix with more details on the datasets, baselines, hyperparameters, etc.
>
> A: In our attached zip file (supplementary material), we included a thorough README with exact commands and hyperparameters to reproduce the results in the paper for all datasets. We will make this explicitly clear in the updated version of Appendix G.

---

### Review · Reviewer_ybg8 · 2024-12-05

**Summary Of Contributions:**

This paper empirically studies results from prior literature on hyperbolic graph representation learning -- that is, embedding of nodes of featurized graphs in hyperbolic spaces.  The paper shows that Euclidean multilayer perceptron implementations as baselines for various recent works are buggy and that, when the bugs are fixed, performance on these baselines improves dramatically, even beyond the performance of state of the art hyperbolic models.

It posits a multi-point explanation for how this state of affairs came about:

1.) Datasets selected for evaluation of hyperbolic GCNs are poorly chosen, in the sense that even methods that ignore graph structure completely perform well on the selected tasks.  This holds even for datasets that perfect $\delta$-hyperbolicity.

2.) The motivation for hyperbolic embedding in the first place is poor -- even if graphs can be embedded in a hyperbolic space with low metric distortion, this is not a justification for assuming that node feature vectors should be as well.

3.) Gromov $\delta$-hyperbolicity is a measure of hyperbolicity that considers only graph structure and not node features.  Thus, this is only a partial measure of appropriateness of hyperbolic embeddings for a given dataset.

The paper then proposes a more rigorous way of benchmarking datasets for analysis of hyperbolic graph learning methods.  It includes the construction of a parametrized family of synthetic datasets, where the parameters control the dependence of node features on graph (in fact, tree) structure.  They show empirically that on this dataset, when parameters are chosen so that graph information is relevant, state-of-the-art hyperbolic GCNs outperform Euclidean GCNs.

**Audience:**

Yes

**Claims And Evidence:**

Yes

**Requested Changes:**

1.) Can the authors provide more intuition regarding the Ollivier-Ricci curvature and how it ought to be interpreted?

2.) Can the authors clarify whether or not the Ollivier-Ricci curvature accounts for node features?

Neither of these are crucial for securing my recommendation for acceptance, but I think that they would improve the paper.

**Strengths And Weaknesses:**

Strengths:

1.) The paper points out a very interesting phenomenon in the literature and explains it at a conceptual level, in addition to showing it with experiments.

2.) The paper provides a dataset whose construction is well-motivated by the conceptual explanation of the flaws of previous work.

3.) The paper lists its main limitations -- one of which is that it does not solve the problem of finding real datasets for which hyperbolic graph learning methods outperform properly implemented Euclidean methods.

Weaknesses:

1.) The paper suggests the use of Ollivier-Ricci curvature profiles in place of $\delta$-hyperbolicity to characterize graph geometry.  I see that this profile can differ between two graphs that are both trees and thus have equal (zero) $\delta$-hyperbolicity.  However, I am not certain of the details regarding the concrete use of the Ollivier-Ricci curvature profile to determine the suitability of a given dataset for analysis via hyperbolic embeddings.  Furthermore, I do not see that this solves the problem of measuring hyperbolicity when node features are involved.

---

> ### Author Response · Authors · 2024-12-23
> **Author Response**
>
> Thank you for the review and the constructive comments. We appreciate that you note our paper “points out a very interesting phenomenon” and explains it on both an experimental and  “conceptual level.” We address your comments from the “Weaknesses” and “Requested Changes” sections below.
>
> ---
> > The paper suggests the use of Ollivier-Ricci curvature profiles in place of delta-hyperbolicity to characterize graph geometry. I see that this profile can differ between two graphs that are both trees and thus have equal (zero) delta-hyperbolicity. However, I am not certain of the details regarding the concrete use of the Ollivier-Ricci curvature profile to determine the suitability of a given dataset for analysis via hyperbolic embeddings. Furthermore, I do not see that this solves the problem of measuring hyperbolicity when node features are involved.
>
> A: You are correct in all of your observations. Our primary purpose in mentioning Ollivier-Ricci curvature was to show that there are more nuanced and rich characterizations of graph geometry, and moreover, that there characterizations offer greater separation between structures that may be viewed as nearly identical by other characterizations (e.g. Gromov $\delta$\-hyperbolicity). Ideally, one could use the curvature profile of a graph to come up with a bespoke embedding space best suited for the graph; in fact, a coarse version of this idea has been attempted by Gu et al. (https://openreview.net/pdf?id=HJxeWnCcF7), but the precise mechanics of such an idea are quite complicated since local geometry is hard to capture with a simple product space. You are also correct about the fact that this does not solve the problem of analyzing the node features together with the graph; one would need to develop a more fundamental set of tools to perform this analysis and we believe this is a very fruitful avenue for future work.
>
> ---
>
> > Can the authors provide more intuition regarding the Ollivier-Ricci curvature and how it ought to be interpreted?
>
> A: Based on the definition given in Appendix C of our paper, one can think of very negative Ollivier-Ricci curvature of an edge resulting from a much higher transport distance between the neighbors of the two does than the actual distance between the nodes. Intuitively, this means the edge is locally something of a “bottleneck” at which a large amount of expansion happens, on either or both sides. On the other hand, if the transport distance is relatively small compared to the distance between the nodes, i.e. the Ollivier-Ricci curvature is either close to zero, this means that the endpoints of the edge have few neighbors and the edge is relatively “flat”; or if the Ollivier-Ricci curvature is positive, the edge is typically included in a number of cycles due to the fact that the neighbors of the two endpoints have non-empty intersection.
>
> ---
>
> > Can the authors clarify whether or not the Ollivier-Ricci curvature accounts for node features?
>
> A: The Ollivier-Ricci curvature does not account for node features and this is a point well worth making. Despite the fact that this is arguably a better tool for analyzing the geometry of graphs than is currently being widely used, it does indeed not take into account node features. One would need to develop a more fundamental set of tools to properly characterize node feature geometry together with the geometry of the graph; we believe this is a very fruitful and important avenue for future work following our paper. We have added this as an explicit clarification to the updated version of the paper.
>
> ---

---

### Author Response · Authors · 2024-12-23
**Author Responses Posted**

Thank you to all the reviewers for their time and their constructive comments! Below each review, we have posted a rebuttal that directly addresses concerns and provides additional detail and context. If you wish to obtain further clarification, please reply in the relevant thread, and we will get back to you as soon as possible.

---

### Decision · Action_Editor_Sm6m · 2025-01-18

**Recommendation:** Accept as is

**Comment:**

The paper was reviewed by three reviewers. The reviewers agreed that the paper's main finding, i.e., that simple Euclidean models outperform or are competitive with hyperbolic models in hyperbolic graph tasks, is surprising and very interesting. They also appreciated the synthetic dataset introduced in the paper on which Euclidean MLP models fail. However, the reviewers also raised concerns mainly about the design of the synthetic dataset, the limited number of real-world datasets, and the strong criticism towards some prior works. The reviewers also asked for more details on the Ollivier-Ricci curvature. The authors comprehensively responded to the reviewers' comments and revised the manuscript accordingly, and all reviewers recommended acceptance of the paper. I believe that the submission now meets the TMLR acceptance criteria, and thus, I am recommending acceptance. I request that the final version of the manuscript implements any remaining items promised in the author feedback.

**Audience:**

Hyperbolic graph learning models have recently become very popular in the graph machine learning community. Therefore, the findings of this paper will be of interest to some individuals in TMLR's audience.

**Claims And Evidence:**

This paper claims that a simple Euclidean MLP model outperforms or matches state-of-the-art hyperbolic neural networks in well-established graph tasks for hyperbolic models. The claim is supported by empirical results in link prediction and node classification tasks. The paper also claims that this surprising discovery is mainly due to issues with the Euclidean baselines used in prior work, faulty modelling assumptions, and the use of improper metrics to quantify geometry of graph datasets. The paper provides evidence to support these claims.

---

> ### Author Response · Authors · 2025-02-15
> **All Requisite Changes Made, Camera-ready Uploaded**
>
> Dear AC,
>
> Thank you for your time and the recommendation! We are happy to have our work accepted at TMLR. We have just finished making the requisite remaining modifications to our paper and have uploaded a camera-ready revision. Additionally, we have made our code publicly available via Github (https://github.com/isaykatsman/Shedding-Light-Hyperbolic-Graph-Learning); this is linked above and directly in Appendix G of our paper. Exact commands for the reproduction of our results are given in the README of this repository. Please let us know if this looks good and/or if you would like any changes to be made.
>
> Best,
> Authors